# Protection Against Rabies Induced by the Non-Replicative Viral Vectors MVA and Ad5 Expressing Rabies Glycoprotein

**DOI:** 10.3390/v17040476

**Published:** 2025-03-27

**Authors:** Debora Patricia Garanzini, Matias Ariel Micucci, Annalies Torres Lopez, Oscar Perez, Gabriela Calamante, Maria Paula Del Medico Zajac

**Affiliations:** 1Instituto de Agrobiotecnología y Biología Molecular (IABiMo), Instituto Nacional de Tecnología Agropecuaria (INTA), Consejo Nacional de Investigaciones Científicas y Técnicas (CONICET), Nicolás Repetto y De Los Reseros S/N, Hurlingham B1686IGC, Buenos Aires, Argentina; debygaranzini@gmail.com (D.P.G.); torres.annalies@inta.gob.ar (A.T.L.); calamante.gabriela@inta.gob.ar (G.C.); 2Servicio de Vacuna Antirrábica (SVAR), Instituto Nacional de Producción de Biológicos, ANLIS-“Dr. Carlos G. Malbrán”, Av. Vélez Sarsfield 563, Ciudad Autónoma de Buenos Aires C1282AFF, Argentina; matias.micucci27@gmail.com (M.A.M.); osky2010@yahoo.com.ar (O.P.)

**Keywords:** rabies, MVA, adenovirus, Ad5, vector-based vaccine, viral vector, glycoprotein

## Abstract

Rabies is a zoonotic viral disease that is preventable through vaccination. Effective control strategies should follow the “One Health” concept, as targeting zoonotic pathogens at their animal source is the most effective and cost-efficient approach to protecting human health. The aim of this study was to develop and evaluate two third-generation anti-rabies vaccines based on non-replicative viral vectors, MVA and Ad5, both expressing rabies virus (RABV) glycoprotein (MVA-RG and Ad-RG). MVA-RG was produced using a platform developed in our laboratory, while Ad-RG was generated using a commercial kit. Protection against rabies was assessed in a mouse intracerebral (IC) RABV challenge model. Our results demonstrated that both vectors provided protection against RABV. MVA-RG and Ad-RG administered in two homologous doses conferred 60% and 60–100% protection against RABV challenge, respectively. The survival rate was influenced by the viral vector, the dose, and the immunization scheme. Remarkably, to our knowledge, our study is the first to report 100% protection against IC RABV challenge using a non-replicative Ad5 in a homologous immunization scheme. These promising results support future evaluation of this vaccine candidate in target animals.

## 1. Introduction

Rabies virus (RABV) is the causative agent of a severe zoonosis that leads to acute encephalitis and death in mammals. This virus belongs to the genus *Lyssavirus* within the family *Rhabdoviridae*, a group of enveloped viruses with a single-stranded negative-sense RNA genome [1]. The viral genome, approximately 12,000 bp in length, encodes five structural proteins: the transmembrane glycoprotein (G), the primary antigen of the virus; the matrix protein (M), located on the inner side of the virion; the nucleoprotein (N), which associates with M and genomic RNA; the phosphoprotein (P or NS); and the polymerase (L) [1]. The transmembrane glycoprotein is the only protein present on the viral surface, contains epitopes recognized by CD4+ and CD8+ T cells, and is the major target of neutralizing antibodies (NAs) [2,3].

Rabies is preventable through vaccination and its control should follow the “One Health” concept. Targeting this zoonotic pathogen at its terrestrial animal source is the most effective and cost-efficient strategy to protect humans. Inactivated-RABV vaccines are widely used for humans, companion animals, and livestock, offering safety and effectiveness. However, they require regular applications to maintain protective immunity in a preventive way, which is expensive and time-consuming. On the other hand, in some countries, baits containing attenuated strains of RABV are used for the vaccination of wild animals (reviewed in [4]). In these cases, it was necessary to demonstrate that these strains do not cause any adverse effects in either the target or non-target species. To overcome these inconveniences, several recombinant viral-vectored vaccines have been developed and assessed for companion animals, wildlife, and livestock, with varying levels of efficacy. However, only a few have been licensed (reviewed in [4,5]).

Two replicative viral-vectored rabies vaccines are commercially available. The first one, obtained over 30 years ago, is based on the vaccinia virus (VACV) Copenhagen strain expressing the glycoprotein of the ERA strain of RABV [6]. Currently commercialized as Raboral V-RG^®^ (Boehringer Ingelheim Animal Health, Ingelheim am Rhein, Germany), this vaccine is used to prevent wildlife rabies in North America and Europe (http://www.raboral.com/, accessed on 20 December 2024). Additionally, a licensed adenovirus-vectored vaccine for wildlife, ONRAB^®^ (Ontario Rabies Vaccine Bait, Artemis Technologies Inc., Guelph, ON, Canada), consists of the replicative viral vector human adenovirus type 5 carrying the gene for the RABV G protein from the ERA strain (AdRG1.3, [7]). This vaccine is currently used in Canada to control rabies in foxes, raccoons, and skunks (https://www.ontario.ca/page/wildlife-rabies-outbreaks-and-control-operations, accessed on 20 December 2024). Furthermore, PUREVAX^®^ (Boehringer Ingelheim) is a non-replicative canarypox virus-based rabies vaccine available for cats.

Non-replicative viral-vectored vaccines are chosen to develop biotechnological vaccines for humans, companion animals, and livestock to ensure safety for humans. Modified vaccinia Ankara virus (MVA) constitutes an excellent platform to develop rationally designed viral-vectored vaccines due to its safety profile, its capability to induce humoral and cellular immune responses, and its intrinsic adjuvant properties [8,9]. Researchers have extensively used MVA as a viral vector for human and veterinary vaccines [10]. Similarly, non-replicative adenoviruses (Ad) are effective vaccine platforms, because they induce potent innate and adaptive immune responses, including antibodies and CD8+ T cell responses. Their advantages include well-established and simple techniques for constructing recombinant adenoviruses, and the ability to achieve high viral titers in vitro [11,12]. While MVA and Ad vectors have been widely assessed as vaccines in preclinical and clinical trials, the first viral-vectored vaccine approved for widespread use in humans was during the COVID-19 pandemic. These vaccines were based on non-replicative Adenovirus 26 (Ad26), replication-defective human adenovirus type 5 (Ad5), or modified chimpanzee adenovirus (ChAdOx1) [13].

To obtain safe and effective third-generation rabies vaccines, in this study, we generated two non-replicative viral vectors based on MVA and Ad5 expressing the RABV glycoprotein (RG). The ability of the recombinant viruses MVA-RG and Ad-RG to protect against RABV was evaluated in a mouse intracerebral challenge model.

## 2. Materials and Methods

### 2.1. Cell Lines and Viral Stocks

Modified vaccinia Ankara virus (MVA) was cultivated in primary chicken embryo fibroblast (CEF) cultures at a low multiplicity of infection (MOI: 0.01). The cells and supernatants were frozen five days after infection. Human adenovirus type 5 ΔE (Ad5) viruses were propagated in human embryonic kidney cells transformed with the E1 region of human adenovirus type 5 (HEK293A). These cells and supernatants were frozen after two days of infection. After two freeze–thaw cycles, viral stocks were clarified to remove cell debris and then ultracentrifuged for virus concentration. Viral pellets were resuspended and ultracentrifuged through a 25% *v*/*v* sucrose cushion. Virus titration was performed using plaque assays in CEFs or HEK293A cells, as described elsewhere [14]. Recombinant MVA carrying the gene encoding the green fluorescent protein (GFP, MVA-GFP) and Ad expressing GFP (Ad-GFP) were provided by Dr. M. Gherardi (Facultad de Medicina, Universidad de Buenos Aires), and Dr. M. Pérez Filgueira (Instituto de Virología, IVIT, UEDD INTA-CONICET), respectively.

The rabies virus strain Challenge Virus Standard (RABV CVS), adapted for propagation in Vero cell culture, was provided by the Servicio de Vacuna Antirrábica (SVAR), ANLIS “Dr. Carlos G. Malbrán”. Virus titration was performed in vivo by inoculating 10-fold dilutions of the viral suspension (30 µL) intracerebrally (IC) into 14–16 g NIH mice. Animal survival was monitored daily for two weeks, and euthanasia was performed upon developing clinical rabies signs. Deaths occurring within the first four days post challenge were considered non-specific. Viral titers were determined using the Reed and Muench method and expressed as 50% lethal dose per mL (LD_50_/mL).

### 2.2. Construction of Recombinant MVA-RG

MVA-RG was generated as previously described [14]. Briefly, the complete coding sequence of RABV glycoprotein (hereafter *RG*) from RABV strain CVS under the regulation of the pE/L synthetic poxviral promoter was obtained from a plasmid available in our laboratory [15] and subcloned into the transfer vector (TV) TV-MTK-GUS-EL [14]. This plasmid also codifies for the β- glucuronidase enzyme (GUS), which facilitates the screening and plaque lysis purification of recombinant MVAs by using its substrate, along with viral genomic regions of the *MVA086R* gene [which codifies for the thymidine kinase (TK) enzyme]. TV-RG was transfected into CEFs previously infected with MVA (MOI: 0.01). Pure recombinant virus (100% blue viral plaques) was achieved after 15 rounds of plaque purification. The presence of the *RG* sequence was confirmed by PCR using primers RG1-RG4 and total DNA from MVA- or MVA-RG-infected cells as templates [15].

The expression of the *RG* sequence was evaluated at the transcriptional level by RT-PCR. Total RNA was extracted from MVA- or MVA-RG-infected cells using TRIzol^TM^ (ThermoFisher Scientific, Waltham, MA, USA), then reverse-transcribed with the reverse transcriptase M-MLVRT (200 U, Promega, Madison, WI, USA) and random hexamers. The resulting cDNA was used to amplify a ~1600 bp product via PCR with RG1 and RG2 primers [15].

### 2.3. Generation of Recombinant Adenoviruses

Recombinant adenoviruses were generated using the ViraPower^TM^ Adenoviral Gateway Expression Kit^TM^ (ThermoFisher Scientific). The complete sequence of *RG* was amplified by PCR from TV-RG using RG1 and RG2 primers [15] and cloned into the entry vector (pCR8^TM^/GW/TOPO^TM^ TA Cloning). An in vitro recombination assay was performed using the obtained plasmid and the destination vector pAd/CMV/V5-DEST^TM^ (a plasmid containing a replication-incompetent Ad5 genome, ∆E1/E3), following the manufacturer’s instructions. HEK293A cells were transfected with pAd/CMV/V5-DEST^TM^-RG using the cationic lipid Lipofectamine 2000 (ThermoFisher Scientific) and incubated until the characteristic adenovirus-cytopathic effect appeared (~10–13 days post transfection).

The presence of the *RG* sequence was determined by PCR using the universal primers T7-promoter (5’TAATACGACTCACTATAGGG) and V5-reverse (5’ACCGAGGAGAGGGTTAGGGAT), which hybridize in sequences present in the destination plasmid, and with total DNA extracted from Ad-RG-infected cells as templates. As described above, the expression of the *RG* sequence was evaluated at the transcriptional level by RT-PCR. Total RNA from uninfected or Ad-RG-infected cells was extracted using TRIzol^TM^ and reverse-transcribed with M-MLVRT using random hexamers. The cDNA obtained was used to amplify a ~900 bp product by PCR with RG1 and RG4 primers [15].

### 2.4. Animals

Female BALB/c (H-2d) mice (6–8 weeks old), certified as specific-pathogen-free, were purchased from the Fundación Facultad de Ciencias Veterinarias (UNLP, La Plata, Argentina) or SVAR. Immunizations were performed in animal facilities at the Instituto de Biotecnología (IABIMO, INTA-CONICET), and the animals were subsequently transferred to the SVAR for the IC RABV challenge. All experiments followed the international welfare guidelines and were approved by the Comité Institucional para el Cuidado y Uso de Animales de Experimentación (CICUAE-CNIA, INTA, Argentina) or in compliance with provision No. 6344/96 of the Administración Nacional de Medicamentos, Alimentos y Productos Sanitarios (ANMAT, Argentina). Mouse suffering was minimized by using isoflurane and CO_2_ inhalation as anesthetic and sacrifice methods, respectively.

### 2.5. Evaluation of Recombinant Viral Vector Efficacy in Homologous and Heterologous Immunization Schemes

Groups of 5–10 mice were immunized with the corresponding immunogen (MVA-RG or Ad-RG) on days 0 and 21. Additionally, control groups received recombinant viruses encoding an unrelated protein (MVA-GFP or Ad-GFP), PBS, or the commercial rabies vaccine VeroRab^®^ (Sanofi Pasteur, Swiftwater, PA, USA). MVA was administered intraperitoneally (IP, 0.8 mL) using 27G needles, while recombinant Ad5 was injected intramuscularly (IM) into the quadriceps (50 μL per leg; 100 μL total) using an insulin syringe (29G). A commercial anti-rabies vaccine (0.25 mL) and PBS (0.8 mL) were administered IP using 27G needles. The commercial vaccine Verorab^®^ (Sanofi Pasteur) was resuspended in 0.5 mL of 0.4% sodium chloride solution, as recommended by the manufacturer. The full dose (0.5 mL) had a potency > 2.5 IU. Mice received 0.25 mL via the IP route.

On days 28–30 post vaccination, the animals were transferred to the SVAR for IC RABV challenge (see below).

### 2.6. Intracerebral RABV Challenge

Fourteen days after the booster, all of the animals were IC challenged with 30 μL of the RABV CVS strain (12.5–50 LD_50_/0.03 mL). Clinical signs and survival were registered daily for two weeks, with euthanasia performed upon the development of clinical signs of rabies. Deaths occurring within the first four days post challenge were considered non-specific. Protection was calculated as the number of surviving animals out of the total number of animals challenged.

## 3. Results

### 3.1. Construction and Molecular Characterization of Recombinant MVA and Ad5 Expressing Rabies Glycoprotein (RG)

Recombinant MVA containing the complete coding sequence of RABV glycoprotein interrupting the *MVA086R* viral gene (TK) was obtained by in vivo homologous recombination between the TV-RG and MVA genome (Figure 1A). Recombinant viruses (hereafter MVA-RG) were isolated by plaque purification based on their capability to produce blue lysis plaques in the presence of X-Gluc (substrate of the GUS marker enzyme, Inalco Pharmaceuthicals, San Luis Obispo, CA, USA). The presence of the *RG* sequence in MVA-RG was confirmed by PCR amplification of a 922 bp fragment (Figure 1B).

Recombinant Ad5 carrying the *RG* sequence (Ad-RG) was generated using the ViraPower^TM^ Adenoviral Gateway Expression Kit^TM^ (ThermoFisher Scientific) (Figure 1D). The inserted nucleotide sequence was confirmed by PCR using T7-promoter and V5-reverse primers that amplify a fragment of approximately 1700 bp (Figure 1E).

The expression of the *RG* sequence by MVA-RG and Ad-RG was determined at the transcriptional level by an RT-PCR assay using RG1/RG2 or RG1/RG4 primers, respectively. Indeed, the amplified fragments, of approximately 1600 and 900 bp, were only observed in samples from cells infected with the recombinant viruses (Figure 1C,F).

### 3.2. Protection Induced by MVA-RG and Ad-RG Against Intracerebral RABV Challenge

To evaluate the level of protection induced by the recombinant viruses in an RABV challenge model, we immunized mice twice with the corresponding dose of the viral vector, following either homologous or heterologous immunization schemes and 14 days after the booster, we performed an intracerebral (IC) challenge with rabies virus.

Firstly, we administered 4 × 10^7^ PFU of MVA-RG twice via the IP route (homologous scheme) or 1 × 10^8^ PFU of Ad-RG via de IM route, followed by a boost with 4 × 10^7^ PFU of MVA-RG (or MVA-GFP) (heterologous scheme). Control groups immunized with MVA-GFP, PBS, or a commercial anti-rabies vaccine were included in the assay (Table 1).

As shown in Table 2 and Figure 2, the homologous MVA-RG scheme induced 60% protection against the IC RABV challenge, whereas the heterologous scheme provided 40% protection. Since both groups primed with Ad-RG showed equivalent survival rates regardless of whether MVA-RG or MVA-GFP was used as the booster, we suggest that survival was primarily due to Ad-RG immunization. As expected, all animals immunized with PBS or MVA-GFP died by day 7 post challenge, whereas those vaccinated with the commercial vaccine survived.

Secondly, we evaluated the protection induced by Ad-RG in homologous prime–boost schemes, administering different doses of the recombinant viral vector (Table 3). The control groups received PBS, a commercial anti-rabies vaccine, or Ad-GFP (a recombinant adenovirus encoding an unrelated protein).

As shown in Table 4 and Figure 3, two immunizations with either 1 × 10^6^ (Group 1) or 1 × 10^7^ PFU (Group 2) of Ad-RG provided 70% and 60% protection, respectively. In contrast, 100% survival was observed in mice injected with 1 × 10^6^ PFU for the prime and 1 × 10^7^ PFU for the boost (Group 3). The control groups responded as expected: all mice in the PBS and Ad-GFP groups died, while those vaccinated with the commercial vaccine showed 100% survival.

These results demonstrate that the MVA-RG and Ad-RG viral vectors induced protection against RABV. The survival rate depends on the viral vector type, administered dose, and immunization scheme.

## 4. Discussion

The rabies vaccines predominantly used worldwide are based on inactivated rabies virus. Additionally, two viral-vectored vaccines are commercially available for wildlife, utilizing replicative strains of vaccinia virus and human adenovirus 5, and one non-replicative vaccine based on canarypox virus for cats [5]. While inactivated vaccines are effective, they require multiple doses to maintain protective immunity, as well a sustained anti-rabies vaccination strategy, in addition to requiring specialized facilities for their production. In contrast, viral-vectored vaccines are also effective against rabies, stimulate both branches of the immune response, and do not involve the manipulation of live rabies virus during manufacturing.

Modified vaccinia Ankara virus (MVA) and human adenovirus type 5 ∆E (Ad5) are non-replicative in mammals and have been extensively evaluated as viral platforms for vaccine development [16].

In this study, we obtained two non-replicative viral vectors expressing the rabies glycoprotein: MVA-RG and Ad-RG. Although glycoprotein expression was confirmed only at the transcriptional level, in vivo protein expression was inferred from the mouse immunization and challenge results. This study assessed the protection induced by these vaccine candidates in a mouse intracerebral RABV challenge model using both homologous and heterologous immunization schemes.

After administering two injections of 4 × 10^7^ PFU of MVA-RG (homologous immunization scheme), we observed 60% protection against RABV challenge. To enhance these protection levels, three strategies are possible: increasing the immunogen dose, extending the time interval between immunizations, or applying a heterologous vaccination scheme.

Regarding the first option, Weyer et al. [17] reported that a single immunization with 1 × 10^8^ PFU of MVA-RG achieved 10% protection in mice, while administering 1 × 10^9^ PFU resulted in 70–80% survival. However, using high doses of recombinant viruses in a mouse model creates challenges for scaling up production for mass vaccination in target animals such as dogs, cats, or livestock.

Another approach involves modifying the time interval between priming and boosting. Administering the booster during the contraction phase of the immune response may prevent any observed increase in the response. Several studies using poxvirus-based vaccines have demonstrated that extending the time interval between the priming and boosting doses improves vaccine-induced immunity [8,18,19,20]. In this context, it would be interesting to test whether increasing the interval between MVA-RG immunizations enhances protection levels against RABV challenge. However, a recent study with the MVA-SARS-2-ST and MVA-SARS-2-S vaccine candidates showed similar immune response levels when comparing 21-day and 56-day intervals between doses [21]. Additionally, the intracerebral RABV challenge applied in this study introduces technical difficulties, as piercing the skull of the animals may increase animal suffering and variability in inoculum delivery.

Heterologous vaccination regimens have demonstrated enhanced immunogenicity and efficacy in humans and animal models by combining adenoviruses and poxviruses in prime–boost schemes [22,23,24,25,26,27,28]

In this study, a heterologous immunization scheme, priming with Ad-RG and boosting with MVA-RG, provided 40% protection against IC RABV challenge. Under the evaluated conditions, this strategy failed to improve protection compared to the homologous scheme. One possible explanation is that the strong immune response elicited by the Ad-RG priming dose (1 × 10^8^ PFU) may have inhibited the effect of MVA-RG. Additionally, as mentioned above, increasing the time interval between immunizations could enhance the specific response. For example, Bruña-Romero et al. [25] reported complete protection against malaria and improved levels of cellular and humoral immune response by reducing the priming dose (from 1 × 10^9^ PFU to 1 × 10^8^ PFU) and extending the time interval before boosting (from 2 to 8 weeks) in the experiments. In their study, the prime and boost consisted of a replication-defective recombinant adenovirus expressing the circumsporozoite (CS) protein of *Plasmodium yoelii* and an attenuated recombinant vaccinia virus expressing the same malaria antigen, VacPyCS, respectively. Unfortunately, this strategy is not easily applicable to the IC challenge model.

The 40% protection after IC RABV challenge observed in the heterologous immunization scheme may be attributed to the administration of a single dose of 1 × 10^8^ PFU of Ad-RG, as the same level of protection was noted after boosting with either MVA encoding the rabies glycoprotein or an unrelated protein (GFP). This protection level is lower than those reported by Zhao et al. [29] and Yan et al. [30]. Their studies showed that a single injection of 1 × 10^8^ GFU (green fluorescent units) of recombinant Ad5, expressing both rabies glycoprotein and canine distemper virus hemagglutinin or severe fever with thrombocytopenia syndrome virus, induces 100% protection against IM RABV challenge.

On the other hand, Kim et al. [2] reported 100% protection following IM immunization with 1 × 10^8^ PFU of Ad5 expressing either the full-length or a truncated version of the RABV glycoprotein after IM challenge, but after IC challenge, 0% to 40% protection was observed. These results emphasize the importance of carefully analyzing efficacy outcomes.

Continuing our study and considering that a single dose of Ad-RG induced 40% protection, it was of great interest to evaluate the efficacy of Ad-RG in a homologous prime–boost scheme. Two injections of 1 × 10^6^ and 1 × 10^7^ PFU resulted in similar protection (70% and 60%, respectively), surpassing the protection induced by a single 1 × 10^8^ PFU dose. Notably, a priming dose of 1 × 10^6^ PFU, followed by a booster dose of 1 × 10^7^ PFU resulted in 100% survival. Although this study did not assess the immune response profile elicited by Ad-RG, these results suggest that priming with 1 × 10^7^ PFU might induce an anti-vector response that inhibits the booster effect on the anti-RABV-specific response.

In this sense, one limitation of Ad-based vaccines is the induction of an anti-vector immune response, which inhibits cell transduction, limits transgene expression, and ultimately reduced overall vaccine effectiveness. This “inhibitory” effect may vary depending on the adenovirus type, dose, and administration route. Pandey et al. [31] analyzed the interference of anti-vector immune responses with antigen-specific responses in BALB/c mice in a prime–boost scheme using different doses of wild-type human adenovirus serotype 5 via the intranasal (IN) route (the natural infection route for adenoviruses) or the IM route (commonly used in Ad-based vaccines). Four weeks later, the mice received a booster dose with different amounts of HAd-HA-NP (HAd expressing the hemagglutinin (HA) and nucleoprotein (NP) of A/Vietnam/1203/04, H5N1 influenza virus) and were subsequently challenged with a non-lethal reassortant influenza virus. The authors observed that higher prime doses led to increased Ad-neutralizing antibody responses, which affected booster dose efficacy. The best protection was achieved with a low prime dose (1 × 10^7^ PFU) followed by a higher boost dose (1 × 10^8^ PFU). In another study, anti-Ad5 immunity was induced by administering different doses of a GFP-recombinant HAd5V (Ad5GFP) via IN or IM routes. Subsequently, mice were immunized subcutaneously (SC) with 1 × 10^9^ PFU/animal of HAd5VCMV. The authors found that only the highest dose of Ad5GFP (1 × 10^7^ PFU) administered via the IM route reduces the CD8+ T cell-specific response against the heterologous antigen (CMV) by 50% [32].

Given the significance of neutralizing antibodies (NA) against Ad5 in reducing its effectiveness for human vaccines development, researchers have focused on the generation of vectors based on animal adenoviruses, particularly chimpanzee-derived Ad vectors (ChAd). While pre-existing NA against simian Ad vectors are relatively rare, potential cross-reactive cellular immunity against Ad should not be overlooked (reviewed by [33]). Several rabies vaccine candidates based on ChAd have demonstrated efficacy against RABV in both mice and non-human primates [34,35]. In recent years, two of these candidates, ChAdOx2-RG and ChAd155-RG, have progressed to phase I human clinical trials, where they induced acceptable anti-RABV NA titers, although at lower levels and shorter durations than a standard inactivated rabies virus vaccine [36,37].

Overall, our results suggest that performing a homologous prime–boost regimen with Ad5 should not be dismissed as a viable vaccine strategy against RABV, as the induction (or lack thereof) of an antigen-specific response will depend on the levels of anti-vector response caused by a prior vaccination.

In this study, we presented the generation of two third-generation rabies vaccine candidates using non-replicative viral vectors in mammals, MVA and Ad5. Both candidates, when administered in homologous vaccination regimens, induced protection against IC RABV challenge. Future research should explore whether increasing the time interval between MVA-RG inoculations enhances protection. Additionally, the promising results of the Ad-RG homologous prime (1 × 10^6^ PFU) - boost (1 × 10^7^ PFU) scheme encourage further evaluation of this vaccine candidate in target animals.

## 5. Conclusions

In the context of developing third-generation vaccines that exclude rabies virus from their formulation while inducing protective immune responses, viral vectors offer an excellent alternative. Moreover, their ability to implement homologous vaccination regimens simplifies their production in industrial systems. Additionally, their safety profile makes them suitable for use in both humans and animals (including companion and production animals), ensuring high levels of protection without the risk of spreading either the infectious agent or the viral vector.

## Figures and Tables

**Figure 1 viruses-17-00476-f001:**
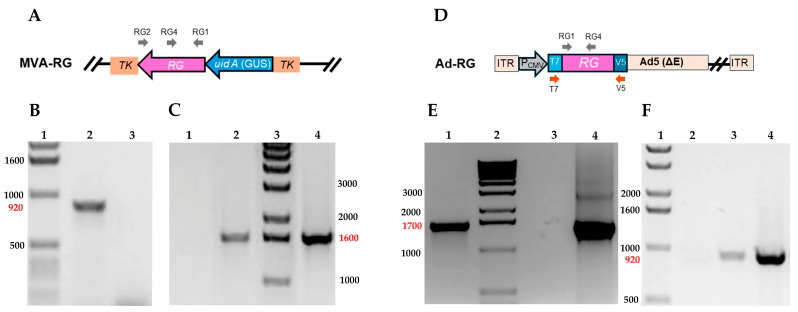
**Molecular characterization of MVA-RG and Ad-RG.** Schematic representation of recombinant MVA (**A**) and Ad-RG (**D**) genomes. Primers used for molecular characterization are indicated with arrows. Presence of *RG* sequence and its transcription was determined by PCR and RT-PCR, respectively, for MVA (**B**,**C**) and Ad (**E**,**F**). (**B**) PCR amplification using RG1/RG4 primers. Molecular weight marker 1 Kb DNA Ladder (Invitrogen, ThermoFisher, line 1); DNA extracted from CEFs infected with MVA-RG (line 2) or MVA (line 3) as templates. (**C**) RT-PCR amplification using RG1/RG2 primers and total RNA extracted from MVA (line 1) or MVA-RG (line 2)-infected CEFs as templates; 1 Kb DNA Ladder (line 3); TV-RG as template (positive control, line 4). (**E**) PCR amplification using T7-promoter and V5-reverse primers. DNA extracted from HEK293A infected with Ad-RG as template (line 1); 1 Kb DNA Ladder (line 2); amplification in absence of DNA (negative control, line 3); pAd/CMV/V5-DEST^TM^-RG as template (positive control, line 4). (**F**) RT-PCR using RG1/RG4 primers. 1 Kb DNA Ladder (line 1); total RNA extracted from non-infected (line 2) or Ad-RG infected (line 3) HEK293A as templates; pAd/CMV/V5-DEST^TM^-RG as template (positive control, line 4). Molecular sizes of DNA ladder are indicated in base pairs (bp) in black. Molecular sizes of amplicons are expressed in bp and shown in red.

**Figure 2 viruses-17-00476-f002:**
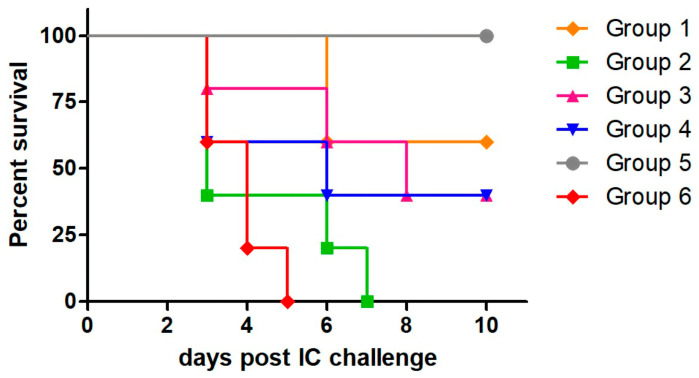
**Protection against RABV induced by MVA-RG in homologous and heterologous immunization schemes.** Mice (n = 5/10) were inoculated with MVA-RG, as indicated in Table 1. Mice from control groups were inoculated with recombinant viruses expressing unrelated protein (MVA-GFP and Ad-GFP), PBS, or commercial anti-rabies vaccine (Verorab^®^). Fourteen days post booster, all groups were IC challenged with 12.5–50 LD_50_ of RABV CVS strain. Survival of mice was checked daily for two weeks. Deaths occurring in first four days after challenge were considered unspecific.

**Figure 3 viruses-17-00476-f003:**
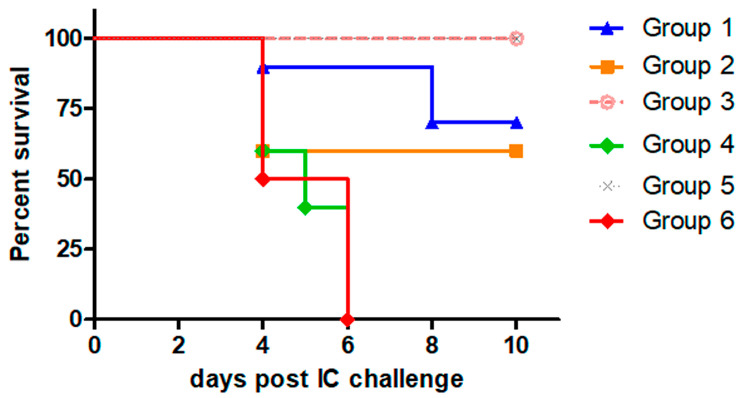
**Survival after IC RABV challenge induced by Ad-RG.** Mice (n = 5/10) were inoculated twice with different doses of Ad-RG, as indicated in Table 2. Mice from control groups were inoculated with Ad-GFP, PBS, or commercial anti-rabies vaccine (Verorab^®^). Fourteen days post booster, all groups were IC challenged with 12.5–50 LD_50_ of RABV CVS strain. Survival of mice was checked daily for two weeks; deaths within first four days after challenge were considered unspecific.

**Table 1 viruses-17-00476-t001:** Immunogen and dose administered to mice in MVA-RG homologous and heterologous schemes.

	Prime	Boost
Group 1	MVA-RG (4 × 10^7^ PFU, IP)	MVA-RG (4 × 10^7^ PFU, IP)
Group 2	MVA-GFP (1 × 10^7^ PFU, IP)	MVA-GFP (1 × 10^7^ PFU, IP)
Group 3	Ad-RG (1 × 10^8^ PFU, IM)	MVA-RG (4 × 10^7^ PFU, IP)
Group 4	Ad-RG (1 × 10^8^ PFU, IM)	MVA-GFP (1 × 10^7^ PFU, IP)
Group 5	Commercial vaccine (0.25 mL, IP)	Commercial vaccine (0.25 mL, IP)
Group 6	PBS (0.8 mL, IP)	PBS (0.8 mL, IP)

**Table 2 viruses-17-00476-t002:** Number of surviving animals and protection after RABV challenge in groups immunized with MVA-RG in homologous and heterologous schemes.

Group	Surviving Animals/Total Animals	Protection After Challenge (%)
1	3/5	60
2	0/5	0
3	2/5	40
4	2/5	40
5	10/10	100
6	0/10	0

**Table 3 viruses-17-00476-t003:** The dose of Ad-RG administered to mice at the prime and the boost.

	Prime	Boost
Group 1	Ad-RG 1 × 10^6^ PFU (IM)	Ad-RG 1 × 10^6^ PFU (IM)
Group 2	Ad-RG 1 × 10^7^ PFU (IM)	Ad-RG 1 × 10^7^ PFU (IM)
Group 3	Ad-RG 1 × 10^6^ PFU (IM)	Ad-RG 1 × 10^7^ PFU (IM)
Group 4	Ad-GFP 1 × 10^6^ PFU (IM)	Ad-GFP 1 × 10^6^ PFU (IM)
Group 5	Commercial vaccine (0.25 mL, IP)	Commercial vaccine (0.25 mL, IP)
Group 6	PBS (0.8 mL, IP)	PBS (0.8 mL, IP)

**Table 4 viruses-17-00476-t004:** Number of surviving animals and protection after RABV challenge in groups immunized with different doses of Ad-RG.

Group	Surviving Animals/Total Animals	Protection After Challenge (%)
1	7/10	70
2	3/5 *	60
3	10/10	100
4	0/5	0
5	10/10	100
6	10/10	0

* This group initially included six animals for the immunization protocol; however, one died two days after challenge, and its death was considered unspecific.

## Data Availability

The data are contained within the article.

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
