# Peer review of "Protection Against Rabies Induced by the Non-Replicative Viral Vectors MVA and Ad5 Expressing Rabies Glycoprotein"

_viruses, 2025, doi:10.3390/v17040476_

Round 1

Reviewer 1 Report (New Reviewer)

Comments and Suggestions for Authors

1          Summary         efficacy in an intracerebral RABV challenge model.    Mouse?

2          Line 47   ‘They are safe and effective but require multiple doses but require regular applications’  Please edit text

3          ‘to maintain immunity in a preventive way,’    meaning: maintain protective immunity?

4          L 80  the first viral-vectored vaccines approved for humans was during Covid-19 pandemic     This is not true, Ebola vector vaccines are being used in man.   Recombivax HB is HBsAg cloned into yeast, not a virus.

5          Please make a clear statement that you are unable to do an in vitro neutralising rabies antibody test, hence the use of MNT.

6          It would be logistically impossible to use a rabies mRNA vaccine as it requires deep freezers for storage.

7          Why has Table I used usual numbering for the groups, but Table 2 uses roman numbering for the same groups? The same applies to Tables 3 and 4.

8          L 284 suggest edit for clarity: ‘do not involve the manipulation of the live rabies virus’ 

 9         L 346 ‘ These results suggest that the IC challenge route is more stringent than the IM route, emphasizing the importance of carefully analyzing efficacy results.’                                                It would be expected that IM challenge provides a more normal challenge route. Less virus would reach the brain, than delivering the virus directly to the brain by IC challenge, in addition to the cerebral trauma inflicted.  Suggest deleting this sentence.

10        An anti-Ad 5 vector antibody is more likely to be found in domestic pets than in mice and so Ad 5 is less suitable as a rabies vector.

Comments on the Quality of English Language

edits suggested

Author Response

Comment 1:Summary, efficacy in an intracerebral RABV challenge model.    Mouse?

Response 1: It is a mouse challenge model. The sentence has been modified.

Comment 2: Line 47, ‘They are safe and effective but require multiple doses but require regular applications’  Please edit text

Response 2: We apologize for the mistake. The sentence has been modified.

Comment 3: ‘to maintain immunity in a preventive way,’    meaning: maintain protective immunity?

Response 3: Yes, we referred to protective immunity. We modified the sentence:

From: Inactivated-RABV vaccines are widely used in humans, companion animals and live-stock. They are safe and effective but require multiple doses but require regular applications to maintain immunity in a preventive way, which is expensive and time consuming.

To: Inactivated-RABV vaccines are widely used for humans, companion animals and live-stock, offering safety and effectiveness. However, they require regular applications to maintain protective immunity in a preventive way, which is expensive and time-consuming..

Comment 4: L 80  the first viral-vectored vaccines approved for humans was during Covid-19 pandemic     This is not true, Ebola vector vaccines are being used in man.   Recombivax HB is HBsAg cloned into yeast, not a virus.

Response 4: In this paragraph, we refer to the first viral-vectored vaccine used massively worldwide. To clarify this, we modified the sentence as follows: 'the first viral-vectored vaccine approved for widespread use in humans was during the Covid-19 pandemic…'

Comment 5: Please make a clear statement that you are unable to do an in vitro neutralising rabies antibody test, hence the use of MNT.

Response 5: All authors of the work formally declare that we currently do not have the technical/methodological capabilities to evaluate rabies virus-neutralizing antibodies in vitro. In this work, we used the IC challenge model as a strategy to select the vaccine candidate and/or immunization regimen that induces the highest protection against RABV challenge. Since we only have an in vivo assay to evaluate RABV-neutralizing antibodies (MNT, which requires a large number of animals) contradicting with the 3Rs principle, the authors decided not to perform the MNT assay at this stage of the evaluation of the vaccine candidates.

Comment 6: It would be logistically impossible to use a rabies mRNA vaccine as it requires deep freezers for storage.

Response 6: We agree with the Reviewer´s comment.

Comment 7:  Why has Table I used usual numbering for the groups, but Table 2 uses roman numbering for the same groups? The same applies to Tables 3 and 4.

Response 7: We apologize for this mistake; the tables have been modified.

Comment 8: L 284 suggest edit for clarity: ‘do not involve the manipulation of the live rabies virus’

Response 8: We agree with the Reviewer´s comment, the word “live” was added.

Comment 9: L 346 ‘ These results suggest that the IC challenge route is more stringent than the IM route, emphasizing the importance of carefully analyzing efficacy results.’                                                It would be expected that IM challenge provides a more normal challenge route. Less virus would reach the brain, than delivering the virus directly to the brain by IC challenge, in addition to the cerebral trauma inflicted.  Suggest deleting this sentence.

Response 9: We agree with the Reviewer's comment and have modified the sentence to: "These results emphasize the importance of carefully analyzing efficacy outcomes."

Comment 10: An anti-Ad 5 vector antibody is more likely to be found in domestic pets than in mice and so Ad 5 is less suitable as a rabies vector.

Response 10: Certainly, when the immunization scheme for the target animals is defined, anti-vector immunity should be considered as an additional factor that could affect the vaccine's efficacy.

Reviewer 2 Report (New Reviewer)

Comments and Suggestions for Authors

Title: Protection against rabies induced by the non-replicative viral vectors MVA and Ad5 expressing rabies glycoprotein.

Authors: Garanzini, D.P., Micucci, M.A., Torres Lopez, A., Perez, O., Calamante, G., and Del Medico Zajac, M.P.

Summary: This paper describes viral vectors using MVA (MVA-RG) and Ad5  (Ad-RG) to express RABV glycoprotein and to evaluate the efficacy of these vectors in a mouse intracerebral RABV challenge model. Both vectors were shown to provide protection against RABV in this mouse model, with he Ad-RG vector approaching 100% protection against RABV intracerebral challenge using the non-replicative Ad5 homologous immunization scheme. Control using the One Health concept at the animal source is the most effective and cost-efficient solution for protecting humans. They will further evaluate these vaccine candidates in target animals.

Revisions needed:

This revised manuscript sufficiently addressed the comments made by previous reviewers.

Author Response

Comment: This revised manuscript sufficiently addressed the comments made by previous reviewers.

Response: We are pleased to have fulfilled the requests of the previous reviewers

This manuscript is a resubmission of an earlier submission. The following is a list of the peer review reports and author responses from that submission.

Round 1

Reviewer 1 Report

Comments and Suggestions for Authors

The authors obtained two variants of viral vectors, MVA and Ad5, which encode the rabies virus glycoprotein (CVS strain). They demonstrated the production of transgene mRNA in cells infected with the obtained viral vectors. Pilot testing of protectiveness using different vaccination regimens was performed. In the experiments conducted, homologous immunization with the Ad5 vector showed the highest efficiency.

Main concerns:

The study does not confirm that the vaccine protein (rabies glycoprotein) is expressed in cells infected with the recombinant viruses (only its RNA is present).

It is not clear how the doses and routes of administration of the recombinant viruses were chosen.

Additional tests are needed to determine the antibodies and neutralizing antibodies to vectors and rabies glycoprotein.

The main problem is the number of animals in the immunization groups, and the presentation of the data from the protection experiments is unclear. There are only a few animals in each group, which is not enough for a reliable comparison. It's also unclear how many animals were in each group at the start. The authors should clarify why they're presenting survival data for a period of 1 to 4 days, even though they say this period of death is non-specific.

The methods used in the study are not detailed enough to allow other scientists to reproduce the study's results.

Technical comments:

Fig. 1: All figures with agarose gel: Please mark all bands on the ladder and the sizes of the target DNA fragments, since not all gels are complete.

The legend for Figure 1 should be corrected. In the legend, "total DNA" and "total RNA" are written, but the gels show PCR products.

Please provide details about how the recombinant viruses were administered (needle, volume).

Line 197: Specify the route of Ad-RG administration.

Table 1 and 2: Information about the commercial vaccine dose and volume is missing.

Please add data on the total number of animals and the number of survived animals in each group.

Please use the same color order or at least similar colors for the controls on Figure 4 as on Figure 2.

Author Response

The authors obtained two variants of viral vectors, MVA and Ad5, which encode the rabies virus glycoprotein (CVS strain). They demonstrated the production of transgene mRNA in cells infected with the obtained viral vectors. Pilot testing of protectiveness using different vaccination regimens was performed. In the experiments conducted, homologous immunization with the Ad5 vector showed the highest efficiency.

Main concerns:

Comment 1: The study does not confirm that the vaccine protein (rabies glycoprotein) is expressed in cells infected with the recombinant viruses (only its RNA is present).

Response 1: In our laboratory, we detected the expression of rabies glycoprotein (RG) in protein extracts from cells infected with MVA-RG using a Western blot assay with a horse polyclonal anti-RABV antibody (as shown in the image below). However, we were unable to detect RG in protein extracts from Ad-RG-infected cells using the available antibodies. Therefore, we confirmed protein expression at the transcriptional level by an RT-PCR assay. Additionally, we believe that in vivo protein expression is demonstrated by the results obtained after mouse immunization and challenge. Indeed, inoculation with MVA-RG or Ad-RG induced anti-RABV antibodies (figures not included in the manuscript but described and discussed below) and conferred protection against RABV challenge, whereas MVA-GFP and Ad-GFP did not.

Since we do not have Western blot data confirming RG expression in Ad-RG-infected cells, we decided not to include the MVA-RG Western blot and instead present the RT-PCR results. However, if the Reviewer considers it beneficial to our manuscript, we can add the Western blot figure and modify the paragraph from the Results section as follows (with added sentences highlighted in yellow):

Expression of rabies glycoprotein in MVA-RG-infected cells was confirmed by Western blot using a horse polyclonal anti-RABV antibody. To detect RG protein in Ad-RG-infected cells, we tested several specific anti-RABV antibodies in Western blot assays. However, these antibodies either failed to recognize RG protein in the cell extracts or exhibited nonspecific interactions that prevented visualization of a distinct RG protein band. Since we were unable to detect rabies glycoprotein, RG sequence expression in MVA-RG- and Ad-RG-infected cells was determined at the transcriptional level by RT-PCR using RG1/RG2 or RG1/RG4 primers, respectively. Amplified fragments of approximately 1600 or 900 bp were observed only in samples from cells infected with the recombinant viruses (Figure 1C and F).

 FIGURE ADDED IN THE ATTACHED FILE

  Rabies glycoprotein detection by Western blot. Line 1 and 2: protein extracts from MVA-GFP or MVA-RG infected cells. Line 3: Thermo Scientific™ PageRuler™ Plus Prestained Protein Ladder. Marker molecular weight (kDa) is indicated in black. The expected molecular weight corresponding to rabies glycoprotein (65 kDa) is indicated in red.

Comment 2: It is not clear how the doses and routes of administration of the recombinant viruses were chosen.

Response 2: MVA-based vaccines are typically evaluated at doses ranging from 1x10⁶ to 1x10⁸ UFP, depending on the antigen and route of immunization. In 2000, Ramirez et al. (1) conducted a detailed study of the immune response (both humoral and cellular) induced by MVA following intraperitoneal (IP) administration of various doses (from 2x10⁶ to 1x10⁸ UFP). Their findings showed that a dose of 2x10⁶ UFP induced the lowest humoral and cellular response against VACV antigens. In our laboratory, we have obtained several MVA-based vaccines targeting viral, bacterial, and parasitic pathogens, and we have evaluated them in mice (2–5). In all cases, we initially assessed the immunogenicity of vaccine candidates by the IP route, as it allows us to compare the immune response induced by recombinant MVA alone (homologous regimen) versus in combination with other immunogens, such as DNA, subunit immunogens, or adenoviruses (heterologous regimens). In fact, when MVA is evaluated in heterologous regimens, it is primarily administered via the IP route.

The dose of 4 × 10⁷ MVA-RG used in our study was determined based on the considerations outlined above, ensuring a viral titer that is scalable for future applications in companion and production animals (as discussed in the manuscript).

Regarding Ad5-based vaccines, the most commonly used route of inoculation is the intramuscular (IM) route, with doses ranging from 1 × 10⁸ to 1 × 10¹⁰ PFU. Specifically, studies using Ad5 as a vaccine candidate against rabies employ a dose of 1 × 10⁸ PFU (6-8). Based on this background, a dose of 1 × 10⁸ PFU of Ad-RG was used in the heterologous prime-boost experiment in our study. However, upon observing an "inhibitory" effect of the Ad-RG-induced response on the boost with MVA, doses 1 and 2 logs lower (1 × 10⁶ and 1 × 10⁷) were tested. Then, based on the results obtained and considering the hypothesis of vector-specific immune response interference, we implemented an immunization scheme using the lowest previously tested dose for the prime, followed by a boost with a one-log increase in the dose.

  1. Ramírez JC, Gherardi MM, Esteban M. Biology of attenuated modified vaccinia virus Ankara recombinant vector in mice: virus fate and activation of B- and T-cell immune responses in comparison with the Western Reserve strain and advantages as a vaccine. J Virol. 2000 Jan;74(2):923-33. doi: 10.1128/jvi.74.2.923-933.2000. PMID: 10623755; PMCID: PMC111613
  2. Ferrer MF, Del Médico Zajac MP et al. Recombinant MVA expressing secreted glycoprotein D of BoHV-1 induces systemic and mucosal immunity in animal models. Viral Immunol 2011;24. https://doi.org/10.1089/vim.2011.0018.
  3. Jaramillo Ortiz JM, Del Médico Zajac MP et al. Vaccine strategies against Babesia bovis based on prime-boost immunizations in mice with modified vaccinia Ankara vector and recombinant proteins. Vaccine. 2014 Aug 6;32(36):4625-32. doi: 10.1016/j.vaccine.2014.06.075.
  4. Morelli MP, Del Medico Zajac MP et al. IL-12 DNA Displays Efficient Adjuvant Effects Improving Immunogenicity of Ag85A in DNA Prime/MVA Boost Immunizations. Frontiers in Cellular and Infection Microbiology, 2020. 23;10:581812. doi: 10.3389/fcimb.2020.581812.
  5. Montenegro VN, Jaramillo-Ortiz JM et al. A prime-boost combination of a three-protein cocktail and multiepitopic MVA as a vaccine against Babesia bigemina elicits neutralizing antibodies and a Th1 cellular immune response in mice. Ticks Tick Borne Dis. 2022; 13(5):101991. doi: 10.1016/j.ttbdis.2022.101991.
  6. Kim, H.H.; Yang, D.K.; Nah, J.J.; Song, J.Y.; Cho, I.S. Comparison of the Protective Efficacy between Single and Combination of Recombinant Adenoviruses Expressing Complete and Truncated Glycoprotein, and Nucleoprotein of the Pathogenic Street Rabies Virus in Mice. Virol J 2017, 14, doi:10.1186/s12985-017-0789-2.
  7. Zhao, Z.; Zheng, W.; Yan, L.; Sun, P.; Xu, T.; Zhu, Y.; Liu, L.; Tian, L.; He, H.; Wei, Y.; et al. Recombinant Human Adenovirus Type 5 Co-Expressing RABV G and SFTSV Gn Induces Protective Immunity Against Rabies Virus and Severe Fever With Thrombocytopenia Syndrome Virus in Mice. Front Microbiol 2020, 11, doi:10.3389/FMICB.2020.01473.
  8. Yan, L.; Zhao, Z.; Xue, X.; Zheng, W.; Xu, T.; Liu, L.; Tian, L.; Wang, X.; He, H.; Zheng, X. A Bivalent Human Adenovirus Type 5 Vaccine Expressing the Rabies Virus Glycoprotein and Canine Distemper Virus Hemagglutinin Protein Confers Protective Immunity in Mice and Foxes. Front Microbiol 2020, 11, doi:10.3389/FMICB.2020.01070.

Comment 3: Additional tests are needed to determine the antibodies and neutralizing antibodies to vectors and rabies glycoprotein.

Response 3: We agree with the Reviewer that additional assays could provide valuable information regarding the performance of our vaccine candidates. In our country, the only assay available to determine RABV-neutralizing antibodies is an in vivo assay that requires the use of 8 animals for each serum dilution. In line with the 3Rs principle (Replacement, Reduction, and Refinement), we plan to select specific time points post-vaccination to minimize animal usage while still obtaining meaningful data.

As a complementary analysis, we tested serum anti-RABV antibodies using an in-house ELISA. For this assay, we used purified inactivated RABV as the antigen and compared absorbance values from animals immunized with MVA-RG or Ad-RG to those from groups immunized with recombinant viruses encoding unrelated proteins (MVA-GFP or Ad-GFP, respectively). Since this assay detects total antibodies against RABV and has not been validated against international standards, we did not include the results in the manuscript. However, we provide the results in this letter and remain at the reviewers’ disposal to include them in the manuscript if deemed necessary.

FIGURE ADDED IN THE ATTACHED FILE

Total anti-RABV antibodies in serum. Serum samples were tested 20 days after the prime (20 dpv) and 13 days after the booster (34 dpv). On the left, data correspond to animals immunized with MVA-RG in homologous and heterologous schemes. On the right, data correspond to animals immunized with different doses of Ad-RG.

Comment 4: The main problem is the number of animals in the immunization groups, and the presentation of the data from the protection experiments is unclear. There are only a few animals in each group, which is not enough for a reliable comparison. It's also unclear how many animals were in each group at the start. The authors should clarify why they're presenting survival data for a period of 1 to 4 days, even though they say this period of death is non-specific.

Response 4: In the Materials and Methods subsection 2.5, Evaluation of Efficacy of Recombinant Viral Vectors in Homologous and Heterologous Immunization Schemes, the number of animals is stated as 5–10 per group, without specifying the exact number in each experimental group. To clarify this and provide more precise results regarding animal survival after challenge, we have added the following tables in the Results subsection 3.2 (Table 2 page 6, Table 4 page 7).

Table 2. Number of surviving animals and protection after RABV challenge in groups immunized with MVA-RG in homologous and heterologous schemes.

Group

Surviving animals/ total animals

% protection after challenge

I

3/5

60

II

0/5

0

III

2/5

40

IV

2/5

40

V

10/10

100

VI

0/10

0

Table 4. Number of surviving animals and protection after RABV challenge in groups immunized with different doses of Ad-RG.

Group

Surviving animals/ total animals

% protection after challenge

I

7/10

70

II

3/5*

60

III

10/10

100

IV

0/5

0

V

10/10

100

VI

10/10

0

*This group initially included six animals during the immunization protocol; however, one died two days after the challenge, and its death was considered unspecific.

After IC challenge, mice are observed for 14 days to monitor clinical signs of rabies, with day 14 marking the end of the experiment. Deaths occurring within the first four days are considered unspecific due to secondary effects of IC inoculation (deep anesthesia, skull injection). Thus, the survival curves represent animal survival over a 10-day period, where day 1 of the curve corresponds to day 5 post-IC challenge. We hope this clarifies the Reviewer's concern

Comment 5: The methods used in the study are not detailed enough to allow other scientists to reproduce the study's results.

Response 5: We would appreciate it if you could indicate which methods are not sufficiently detailed. We did not provide details on the production and purification methods of MVA/Ad5-vectored vaccines, as we consider them commonly used; however, this can be modified if required. Additionally, we have added information to some sections of the methods. Below, you can find the modified texts highlighted in yellow.

2.1. Cell lines and viral stocks.

Modified vaccinia Ankara virus (MVA) was cultivated in primary cultures of chicken embryo fibroblasts (CEFs) at a low multiplicity of infection (0.01) and cells and supernatants were frozen 5 days post infection. Human adenovirus type 5 ΔE (Ad5) viruses were propagated in human embryonic kidney cells transformed with the E1 region of human adenovirus type 5 (HEK293A) and cells and supernatants were frozen 2 days post infection. After two cycles of freezing and thawing, viral stocks were clarified to eliminate cell debris and virus concentration was made by ultracentrifugation. Viral pellets were resuspended and ultracentrifuged through a 25% v/v sucrose cushion Titration of viral stocks was made by plaque assay in CEFs or HEK293A cells, as described elsewhere [14]. Viral titters were expressed as plaque forming units / mL (PFU/mL). Recombinant MVA carrying the gene encoding the green fluorescent protein (GFP, MVA-GFP), was obtained previously in our laboratory (unpublished results) and Ad5 expressing GFP (Ad-GFP) was provided by Dr. M. Pérez Filgueira (Instituto de Virología, IVIT, UEDD INTA-CONICET).

Rabies virus strain Challenge Virus Standard (RABV CVS), adapted for propagation in Vero cell culture, was provided by the Servicio de Vacuna Antirrábica (SVAR), ANLIS "Dr. Carlos G. Malbrán". Virus titration was performed in vivo by inoculating 10-fold dilutions of the viral suspension (30 µL) into 14–16 g NIH mice via the intracerebral (IC) route. Animal survival was monitored daily for two weeks, and animals were euthanized upon developing clinical signs of rabies. Deaths occurring within the first four days post-challenge were considered non-specific. The viral titer was determined using the Reed and Muench method and expressed as 50% lethal dose per mL (LD₅₀/mL).

2.2. Construction of recombinant MVA-RG.

MVA-RG was generated as described before [14]. Briefly, the complete coding sequence of RABV glycoprotein (hereafter named RG) from RABV strain CVS under the regulation of the pE/L synthetic poxviral promoter was obtained from a plasmid available in our laboratory [15] and subcloned into the transfer vector (TV) TV-MTK-GUS-EL [14]. This plasmid also codifies for the β- glucuronidase enzyme (GUS), that facilitates screening and plaque lysis purification of recombinant MVAs by using its substrate, and the viral genomic regions of the MVA086R gene [which codifies for the thymidine kinase (TK) enzyme]. TV-RG was transfected into CEFs previously infected with MVA (with a MOI 0.01), and pure recombinant virus (100% of blue viral plaques) was achieved after 15 plaque-purification rounds. The presence of the RG sequence was confirmed by PCR using primers RG1-RG4 and total DNA from MVA or MVA-RG infected cells as template [15].

Expression of RG sequence was evaluated at the transcriptional level by RT-PCR. Total RNA from MVA or MVA-RG infected cells was extracted using TRIzolTM (ThermoFisher Scientific) and reverse-transcribed with the reverse transcriptase M-MLVRT (200 U, Promega) using random hexamers. The cDNA obtained was used to amplify by PCR a product of approximately 1600 bp with primers RG1 and RG2 [15].

2.3. Generation of Recombinant Adenoviruses.

The ViraPower™ Adenoviral Gateway Expression Kit™ (ThermoFisher Scientific) system was employed for the generation of recombinant adenoviruses. The complete sequence of RG was amplified by PCR from TV-RG using RG1 and RG2 primers [15] and cloned into the entry vector (pCR8TM/GW/TOPOTM TA Cloning) and then an in vitro recombination assay was performed using this plasmid and the destination vector (pAd/CMV/V5-DEST™, containing replication incompetent Ad5 genome, ∆E1/E3) following the manufacturer's instructions. Transfection of HEK293A cells was performed using the cationic lipid Lipofectamine 2000 (ThermoFisher Scientific) and incubated until characteristic cytopathic effect of adenovirus was observed (around 10-13 days post-transfection). The presence of the RG sequence was deter-mined by PCR using universal primers T7-promoter (5´TAATACGACTCACTATAGGG) and V5-reverse (5´ACCGAGGAGAGGGTTAGGGAT), which hybridizes in sequences present in the destination plasmid, and total DNA extracted from Ad-RG infected cells as template. As described above, expression of RG sequence was evaluated at the transcriptional level by RT-PCR. Total RNA from uninfected or Ad-RG infected cells was extracted using TRIzolTM and reverse-transcribed with M-MLVRT using random hexamers. The cDNA obtained was used to amplify by PCR a product of approximately 900 bp with primers RG1 and RG4 [15]

2.5. Evaluation of efficacy of recombinant viral vectors in homologous and heterologous immunization schemes.

Groups of 5-10 animals were immunized with the corresponding immunogen (MVA-RG or Ad-RG) on days 0 and 21 as detailed in Table 1 and 2. Additionally, mice of control groups were injected with recombinant viruses encoding a non-related protein (MVA-GFP or Ad-GFP), PBS or the commercial rabies vaccine VeroRab (Sanofi Pasteur). MVA was administered via the intraperitoneal (IP) route (0.8 mL) using 27G needles. Recombinant Ad5 was injected via the intramuscular (IM) route into the quadriceps, administering 50 μL in each leg (100 μL total volume) using an insulin syringe (29G). A commercial anti-rabies vaccine (0.25 mL) and PBS (0.8 mL) were administered via the IP route using 27G needles. The commercial vaccine Verorab® (Sanofi Pasteur) was resuspended in 0.5 mL of 0.4% sodium chloride solution, as recommended by the manufacturer. The full dose (0.5 mL) has a potency > 2.5 IU. Mice received 0.25 mL via the IP route. On day 28-30 post vaccination, animals were moved to the SVAR to perform the RABV IC challenge (see below).

Technical comments:

Comment 6: Fig. 1: All figures with agarose gel: Please mark all bands on the ladder and the sizes of the target DNA fragments, since not all gels are complete.

Response 6: We addressed this issue.

Comment 7: The legend for Figure 1 should be corrected. In the legend, "total DNA" and "total RNA" are written, but the gels show PCR products.

Response 7: The legend has been modified as follows:  

Figure 1. Molecular characterization of MVA-RG and Ad-RG. Schematic representation recombinant MVA (A) or Ad-RG (D) genome. Primers used for molecular characterization are indicated with arrows. The presence of RG sequence was determined by PCR amplification and the corroboration of expression of RG RNA by RT-PCR for MVA (B, C) or Ad (E, F). (B) PCR amplification using RG1 and RG4 primers. Line 1: molecular weight marker 1 Kb DNA Ladder (Invitrogen, ThermoFisher); amplification in the presence of DNA extracted from CEFs infected with MVA-RG (line 2) or MVA (line 3). (C) RT-PCR amplification using RG1/RG2 primers and total RNA extracted from MVA (line 1) or MVA-RG (line 2) infected CEFs; line 3: 1 Kb DNA Ladder; line 4: PCR positive control using TV-RG as template. (E) PCR amplification using T7-promoter and V5-reverse primers. Amplification in the presence of DNA extracted from HEK293A infected with Ad-RG (line 1); molecular weight marker 1 Kb DNA Ladder (line 2); amplification in the absence of DNA (line 3); PCR-positive control: amplification in the presence of pAd/CMV/V5-DEST™-RG (line 4). (F) Line 1: 1 Kb DNA Ladder; RT-PCR using RG1/RG4 primers and total RNA extracted from non-infected (line 2) or Ad-RG infected (line 3) HEK293A; line 4: PCR positive control using pAd/CMV/V5-DEST™-RG as template. The molecular sizes of the DNA ladder are indicated in base pairs (bp) in black. The molecular sizes of the amplicons are expressed in bp and shown in red.

Comment 8: Please provide details about how the recombinant viruses were administered (needle, volume).

Response 8: The following information was included in Materials and Methods subsection 2.5., Evaluation of efficacy of recombinant viral vectors in homologous and heterologous immunization schemes.

MVA was administered via the IP route (0.8 mL) using 27G needles. Recombinant Ad5 was injected via the IM route into the quadriceps, administering 50 μL in each leg (100 μL total volume) using an insulin syringe (29G). A commercial anti-rabies vaccine (0.25 mL) and PBS (0.8 mL) were administered via the IP route using 27G needles.

Comment 9: Line 197: Specify the route of Ad-RG administration.

Response 9: The route of administration (IM) has been added to the text.

Comment 10: Table 1 and 2: Information about the commercial vaccine dose and volume is missing.

Response 10:  The commercial vaccine Verorab® (Sanofi Pasteur) was resuspended in 0.5 mL of 0.4% sodium chloride solution, as recommended by the manufacturer. The full dose (0.5 mL) has a potency > 2.5 IU. Mice received 0.25 mL via the IP route.

This information was added in Materials and Methods subsection 2.5., Evaluation of efficacy of recombinant viral vectors in homologous and heterologous immunization schemes, and the volume of vaccine administered was included in Tables 1 and 3.

Comment 11: Total number of animals and the number of survived animals in each group.

Response 11: We have included two tables (Table 2 and 4) in the revised version of the manuscript to present the requested data.

Comment 12: Please use the same color order or at least similar colors for the controls on Figure 4 as on Figure 2.

Response 12: We modified the graphs by using the same colors for the control groups. We hope this facilitates their interpretation.

Reviewer 2 Report

Comments and Suggestions for Authors

In this article, the authors report the evaluation of two recombinant rabies vaccines using non-replicating vectors in a challenge model in mice.

General comments:

The authors are exploring the efficacy of MVA and replication incompetent Ad5 vectors in rabies.

The characterization of the constructs is minimal and not complete. The controls performed by the authors do not guarantee a proper expression of the rabies glycoprotein. It is a limitation of the study.

The authors use the IC rabies challenge model (NIH test). This test could be used for a proof of concept of the vaccine induced protection. However, one could challenge its use for the optimization of a prime-boost. Why didn’t the authors measure neutralizing antibodies in mice? It could have been an interesting piece of information to compare the immunogenicity between the vaccine regimen. It is another limitation of the study.

Finally, there are commercial inactivated vaccines which provide a full and long-lasting protection after a single dose. What is the rationale to design prime-boost approaches which will make vaccine manufacturing and vaccination more complex and expensive?

Detailed comments:

Line 38: syntax error, review the sentence (“always is associated…”)

Line 45: controlling the terrestrial sources of rabies may be easier than other sources like bats.

Line 47: in the veterinary segment, some adjuvanted inactivated rabies vaccines are protective after one dose and can even provide 3-year duration of immunity after a single dose (efficacy demonstrated by IC challenge in the target species). This sentence should be corrected.

Lines 49-50: existing rabies vaccines do not cause severe adverse events. The authors should explain what the problem is.

Lines 114 & 131: why didn’t the authors use the same primers for the amplification of the cDNA fragment in MVA and Ad infected cells to check expression at transcription step? One would imagine that you could have amplified the same rabies glycoprotein G sequence in both cases.

Sections 2.2 & 2.3: the authors checked the expression of the antigen at transcriptional level. They didn’t check the quality of the expressed glycoprotein. The controls are then very minimal and cannot rule out any problem at translational level. The authors should comment on this and at least mention in the discussion that it is a limitation of their study.

Sections 2.5 & 2.6: the IC rabies challenge model (NIH test) has been designed and calibrated to test inactivated rabies vaccines. Commercial recombinant viral rabies vaccines are not released on the basis of NIH test. In addition, in the context of the 3R’s, NIH test tends to be replaced by serology in mice or even better by in vitro potency tests. The authors should justify the use of this model to assess the immunogenicity of their vaccines.

Lines 239-240: same comment as for line 47. Inactivated rabies vaccines can be very potent and provide long-term protection after a single administration.

Lines 244-246: these vectors have indeed been extensively studies for various targets in different species. One of the drawbacks of the adenovirus vector is the interference of anti-vector immunity on the efficacy of boosters. The authors may want to comment on this.

Lines 273-275: there is indeed a lot of literature on heterologous prime-boost protocols. They are usually applied to difficult pathogens requiring a strong T-cell immunity (e.g. HIV; malaria; …). For a strong immunogen like rabies, there is ample evidence that a single shot of an inactivated vaccine or canarypox-vectored vaccine can protect. Heterologous prime-boost come with a complexity in development, manufacturing and supply, and is probably not the best approach for rabies vaccine.  

Lines 289-290: since neutralizing antibodies are the key driver of protection against rabies, it would have made sense to follow the antibody responses after different vaccine protocols to get a better understanding of their immunogenicity. The IC challenge could have been done in a second instance.

Lines 302-303: The NIH rabies challenge model has been developed and calibrated to test inactivated rabies vaccines in veterinary and human segments. This test is probably not the best one to evaluate and compare recombinant vaccines and vaccination protocols.

Lines 310-313: see previous comment on adenovirus vector. The authors’ observation corroborates the anti-vector immunity and its interference on booster efficacy. This would also apply on subsequent vaccine boosters and may be a concern in the context of vaccination schemes requiring re-vaccination.

Lines 345-348: as previously commented, for a pathogen like rabies, vaccination protocol should remain simple. Developing different types of adenovirus vectors for heterologous prime-boost regimen or using different doses of the same adenovirus in a homologous prime-boost approach complicate the logistics of vaccination. As written by the authors in the conclusion, the industrial aspects must be taken into consideration (simplicity, supply-chain, cost, …)

Lines 349-356: the development of recombinant rabies vaccine is more complex than just a test in mice. The authors should replace the word “development”. As already written, the mice IC challenge is probably not the best model to optimize the MVA and/or Ad prime-boost. Shouldn’t this optimization be done by serology in the target species instead of IC challenge in mice? The authors should comment on this.

Lines 357-364 (conclusion): viral vectors are indeed an attractive platform. However, mRNA could be another one.

Author Response

Point by point response.

Reviewer 2

In this article, the authors report the evaluation of two recombinant rabies vaccines using non-replicating vectors in a challenge model in mice.

General comments:

Comment 1: The authors are exploring the efficacy of MVA and replication incompetent Ad5 vectors in rabies.

The characterization of the constructs is minimal and not complete. The controls performed by the authors do not guarantee a proper expression of the rabies glycoprotein. It is a limitation of the study.

Response 1: In our laboratory, we confirmed the expression of rabies glycoprotein (RG) in protein extracts from MVA-RG-infected cells using a Western blot assay with a horse polyclonal anti-RABV antibody (as shown in the image below). However, despite testing the available antibodies, we were unable to detect RG in protein extracts from Ad-RG-infected cells. To verify protein expression, we instead assessed RG transcription through an RT-PCR assay.

FIGURE ADDED IN THE ATTACHED FILE

Rabies glycoprotein detection by Western blot. Line 1 and 2: protein extracts from MVA-GFP or MVA-RG infected cells. Line 3: Thermo Scientific™ PageRuler™ Plus Prestained Protein Ladder. Marker molecular weight (kDa) is indicated in black. The expected molecular weight corresponding to rabies glycoprotein (65 kDa) is indicated in red.

As we lack Western blot data confirming RG expression in Ad-RG-infected cells, we opted not to include the MVA-RG Western blot and instead present the RT-PCR results. However, if the Reviewer believes it would improve our manuscript, we can incorporate the Western blot figure and revise the paragraph in the Results section accordingly (with added sentences highlighted in yellow).

Expression of rabies glycoprotein in MVA-RG-infected cells was confirmed by Western blot using a horse polyclonal anti-RABV antibody. To detect RG protein in Ad-RG-infected cells, we tested several specific anti-RABV antibodies in Western blot assays. However, these antibodies either failed to recognize RG protein in the cell extracts or exhibited nonspecific interactions that prevented visualization of a distinct RG protein band. Since we were unable to detect RG protein, RG sequence expression in MVA-RG- and Ad-RG-infected cells was determined at the transcriptional level by RT-PCR using RG1/RG2 or RG1/RG4 primers, respectively. Amplified fragments of approximately 1600 or 900 bp were observed only in samples from cells infected with the recombinant viruses (Figure 1C and F).

Moreover, it is important to mention that the immune response is induced in vivo by the de novo synthesis of RG from viral vectors. Consequently, since we have a mouse challenge model for rabies, we focus more on evaluating protection rather than the level of RG protein expression in vitro.

Our research group has been working with recombinant viral vectors for the past 20 years, and based on our experience, it is more important to evaluate the in vivo immunogenicity and protective capacity of vaccine candidates than to characterize the level of recombinant protein expression in vitro. We consider RT-PCR as a suitable approach when the recombinant protein is not easily detected by Western blot.

Comment 2: The authors use the IC rabies challenge model (NIH test). This test could be used for a proof of concept of the vaccine induced protection. However, one could challenge its use for the optimization of a prime-boost. Why didn’t the authors measure neutralizing antibodies in mice? It could have been an interesting piece of information to compare the immunogenicity between the vaccine regimen. It is another limitation of the study.

Response 2: We agree with the reviewer that having data on the immune response induced by each vaccine candidate and immunization scheme would be highly valuable.

However, in this study, we used the IC challenge model as a strategy to select the vaccine candidate and/or immunization regimen that induces the highest protection against RABV challenge. Currently, we only have an in vivo assay to evaluate RABV-neutralizing antibodies, which requires a large number of animals, contradicting with the 3Rs principle. In the future, our goal is to study the humoral and cellular immune response profile induced by the viral vector-based vaccination regimen that provided the highest protection against RABV. To achieve this, we will work on implementing various in vitro techniques, including the seroneutralization assay.

As an additional analysis, we evaluated serum anti-RABV antibodies using an in-house ELISA. For this assay, we employed purified inactivated RABV as the antigen and compared the absorbance values of animals immunized with MVA-RG or Ad-RG to those of groups immunized with recombinant viruses expressing unrelated proteins (MVA-GFP or Ad-GFP, respectively). Since this assay has not been validated against international standards, we did not include the results in the manuscript. However, we present the findings in this letter and are available to incorporate them into the manuscript if the reviewers find it necessary.

FIGURE ADDED IN THE ATTACHED FILE

Total anti-RABV antibodies in serum. Serum samples were tested 20 days after the prime (20 dpv) and 13 days after the booster (34 dpv). On the left, data correspond to animals immunized with MVA-RG in homologous and heterologous schemes. On the right, data correspond to animals immunized with different doses of Ad-RG.

Comment 3: Finally, there are commercial inactivated vaccines which provide a full and long-lasting protection after a single dose. What is the rationale to design prime-boost approaches which will make vaccine manufacturing and vaccination more complex and expensive?

Response 3: The goal of our laboratory is to generate and evaluate new rabies immunogens that do not require the pathogen for their production. This approach would eliminate the costs associated with high-biosafety facilities and the maintenance of immunity in personnel involved in manufacturing.

We agree with the Reviewer that implementing a prime-boost regimen would increase costs; however, we believe our results provide valuable insights into the efficacy of third-generation rabies vaccines, helping to identify optimal doses and immunization strategies.

Detailed comments:

Comment 4:  Line 38: syntax error, review the sentence (“always is associated…”)

Response 4: This mistake has been corrected.

Comment 5: Line 45: controlling the terrestrial sources of rabies may be easier than other sources like bats.

Response 5: The sentence has been modified as follows, with changes highlighted in yellow.

Indeed, controlling this zoonotic pathogen at its terrestrial animal source is the most effective and cost-efficient solution for protecting humans

Comment 6: Line 47: in the veterinary segment, some adjuvanted inactivated rabies vaccines are protective after one dose and can even provide 3-year duration of immunity after a single dose (efficacy demonstrated by IC challenge in the target species). This sentence should be corrected.

Response 6: We agree with the Reviewer. We have modified that sentence as follows (suggested by the Academic Editor):

They are safe and effective but require regular applications to maintain immunity in a preventive way, which is expensive and time consuming.

Comment 7: Lines 49-50: existing rabies vaccines do not cause severe adverse events. The authors should explain what the problem is.

Response 7: We apologize for the confusion. What we meant to say is that, before being approved for use in baits, their safety must be demonstrated. Obviously, the strains currently in use are safe; that is not in question. We have reworded the sentence as follows:

In these cases, it was necessary to demonstrate that these strains do not cause any adverse effects in either target or non-target species.

Comment 8: Lines 114 & 131: why didn’t the authors use the same primers for the amplification of the cDNA fragment in MVA and Ad infected cells to check expression at transcription step? One would imagine that you could have amplified the same rabies glycoprotein G sequence in both cases.

Response 8: In our lab, we routinely use two sets of primers (RG1/RG2 and RG1/RG4) to amplify the RG sequence by PCR. Unfortunately, we cannot determine whether the issue in this study was purely technical (e.g., the quality of the RNA or cDNA obtained for RT-PCR of the full sequence) or due to the operator's choice. The generation and molecular characterization of each viral-vectored vaccine candidate were performed in different years by different participants in this study, who may have selected one primer set over the other when performing PCR.

Comment 9: Sections 2.2 & 2.3: the authors checked the expression of the antigen at transcriptional level. They didn’t check the quality of the expressed glycoprotein. The controls are then very minimal and cannot rule out any problem at translational level. The authors should comment on this and at least mention in the discussion that it is a limitation of their study.

Response 9: As mentioned at the beginning of this letter, using the anti-RABV antibodies available in our laboratory, we were only able to detect the RABV glycoprotein in the extract of cells infected with MVA-RG. For this reason, we present the RT-PCR results for both recombinant viruses.

As requested by the reviewer, we modified a paragraph in the Discussion as follows (changes highlighted in yellow):

In this study, we obtained two non-replicative viral vectors expressing the rabies glycoprotein, MVA-RG and Ad-RG. Although glycoprotein expression could only be verified at the transcriptional level, we consider that in vivo protein expression could be demonstrated by the results obtained after mouse immunization and challenge. Thus, we investigated the protection induced by these vaccine candidates in a mouse RABV intracerebral challenge model, using both homologous and heterologous immunization schemes.

Comment 10: Sections 2.5 & 2.6: the IC rabies challenge model (NIH test) has been designed and calibrated to test inactivated rabies vaccines. Commercial recombinant viral rabies vaccines are not released on the basis of NIH test. In addition, in the context of the 3R’s, NIH test tends to be replaced by serology in mice or even better by in vitro potency tests. The authors should justify the use of this model to assess the immunogenicity of their vaccines.

Response 10: In our study, we used IC challenge as an efficacy test, based on the NIH challenge protocol. In this protocol, inactivated vaccines are administered on days 0 and 7, and animals are challenged IC at 14 days post-vaccination. We followed a vaccination schedule commonly used for recombinant vaccines, immunizing on days 0 and 21 and performing the challenge 14 days later.

We agree that in vitro tests that minimize the use of experimental animals are preferable; however, in our case, we only have in vivo techniques available to assess neutralizing antibodies. For this reason, we hope that in the future we could evaluate the neutralizing antibody response of selected relevant time points after immunization.

Comment 11: Lines 239-240: same comment as for line 47. Inactivated rabies vaccines can be very potent and provide long-term protection after a single administration.

Response 11: The sentence was modified as highlighted in yellow:

While inactivated vaccines are effective, they require regular revaccinations to maintain protective immunity and necessitate specialized facilities for their production

Comment 12: Lines 244-246: these vectors have indeed been extensively studies for various targets in different species. One of the drawbacks of the adenovirus vector is the interference of anti-vector immunity on the efficacy of boosters. The authors may want to comment on this.

Response 12: We addressed this issue in the Discussion section, starting on line 318 of the manuscript. Specifically, stated that ‘This “inhibitory effect may vary depending on the types of adenoviruses, the dose administered, and the route of administration.’ Additionally, we cited and discussed two studies that examined the anti-vector effect on both the immunogenicity and efficacy of Ad5 vectors (1, 2). The authors reported that only high doses administered at prime (1 × 10⁷–1 × 10⁹ PFU) could interfere with the booster effect. we

  • Pandey, A.; Singh, N.; Vemula, S. V.; Couëtil, L.; Katz, J.M.; Donis, R.; Sambhara, S.; Mittal, S.K. Impact of Preexisting Adenovirus Vector Immunity on Immunogenicity and Protection Conferred with an Adenovirus-Based H5N1 Influenza Vaccine. PLoS One 2012, 7, doi:10.1371/journal.pone.0033428.
  • De Andrade Pereira, B.; Bouillet, L.E.M.; Dorigo, N.A.; Fraefel, C.; Bruna-Romero, O. Adenovirus Specific Pre-Immunity Induced by Natural Route of Infection Does Not Impair Transduction by Adenoviral Vaccine Vectors in Mice. PLoS One 2015, 10, doi:10.1371/journal.pone.0145260.

Comment 13: Lines 273-275: there is indeed a lot of literature on heterologous prime-boost protocols. They are usually applied to difficult pathogens requiring a strong T-cell immunity (e.g. HIV; malaria; …). For a strong immunogen like rabies, there is ample evidence that a single shot of an inactivated vaccine or canarypox-vectored vaccine can protect. Heterologous prime-boost come with a complexity in development, manufacturing and supply, and is probably not the best approach for rabies vaccine. 

Response 13: We agree with the Reviewer that heterologous prime-boost strategies are more expensive than other available alternatives for rabies control. In our study, we evaluated this strategy solely to determine whether it improved the protection conferred by MVA-RG or if it was definitively not a good candidate for rabies control. Based on the results obtained, it is not a good option to test it in target animals.

Comment 14: Lines 289-290: since neutralizing antibodies are the key driver of protection against rabies, it would have made sense to follow the antibody responses after different vaccine protocols to get a better understanding of their immunogenicity. The IC challenge could have been done in a second instance.

Response 14: We agree with the Reviewer but at this moment, we only have an in vivo assay to measure neutralizing antibodies, which requires a large number of animals for the determinations (8 animals per serum dilution).

As previously mentioned, our laboratory has an in-house ELISA that has not been validated with international standards, which gives us an idea of the total anti-RABV antibody levels in serum. However, since this assay measures total antibodies and is not validated, the results are only indicative. In consequence, we selected the vaccine candidate based on the protection induced after IC challenge.

Comment 15: Lines 302-303: The NIH rabies challenge model has been developed and calibrated to test inactivated rabies vaccines in veterinary and human segments. This test is probably not the best one to evaluate and compare recombinant vaccines and vaccination protocols.

Response 15: As mentioned earlier, we used a modified protocol of the NIH rabies challenge model. We implemented a vaccination schedule commonly used for recombinant vaccines, with immunizations on days 0 and 21, followed by the challenge 14 days later. Since we only have in vivo assays to evaluate RABV neutralizing antibodies and efficacy, we chose this approach to identify the best candidates for further evaluation.

Comment 16: Lines 310-313: see previous comment on adenovirus vector. The authors’ observation corroborates the anti-vector immunity and its interference on booster efficacy. This would also apply on subsequent vaccine boosters and may be a concern in the context of vaccination schemes requiring re-vaccination.

Response 16: Anti-vector response is a critical factor to consider when proposing homologous vaccination regimens. In our study, we observed that administering different doses in the prime and boost may help avoid this issue, as 100% protection was achieved.

We believe that this approach should also be evaluated in target species, along with the duration of the immune response, which will indicate when revaccination is needed.

Comment  17: Lines 345-348: as previously commented, for a pathogen like rabies, vaccination protocol should remain simple. Developing different types of adenovirus vectors for heterologous prime-boost regimen or using different doses of the same adenovirus in a homologous prime-boost approach complicate the logistics of vaccination. As written by the authors in the conclusion, the industrial aspects must be taken into consideration (simplicity, supply-chain, cost, …)

Response 17: We agree with the Reviewer's comment. We believe that when selecting a vaccine candidate, practical application and economic aspects must be taken into account.

In lines 370-378, we discussed the results and progress achieved with simian or chimpanzee adenoviruses as rabies vaccine candidates. However, it was not our intention to propose heterologous regimens combining different adenoviruses, as this would significantly complicate vaccination logistics.

We believe it is essential to evaluate the duration of the immune response induced by our candidates to ultimately establish vaccination regimens that simplify vaccine administration logistics. In the future, our goal is to assess the induction of the anti-RABV response in target species and its duration.

Comment 18: Lines 349-356: the development of recombinant rabies vaccine is more complex than just a test in mice. The authors should replace the word “development”. As already written, the mice IC challenge is probably not the best model to optimize the MVA and/or Ad prime-boost. Shouldn’t this optimization be done by serology in the target species instead of IC challenge in mice? The authors should comment on this.

Response 18: The reviewer's comment was taken into account, and the word "development" was replaced with "generation". In our study, we used the mouse challenge model to select the best third-generation vaccine candidate to be tested in target species. Unfortunately, in our country, we do not have in vitro assays available to evaluate neutralizing antibody levels. The SVAR-Malbrán Institute (co-authors of this manuscript), conducts an in vivo assay that requires a large number of animals. Finally, our future goal is to evaluate some of the immunization schemes presented in this study in the target species. However, it is necessary to implement in vitro techniques to determine this serological response.

Comment 19: Lines 357-364 (conclusion): viral vectors are indeed an attractive platform. However, mRNA could be another one.

Response  19: We agree with this comment. It is important to mention that during last year’s RITA conference, some results were presented on the development of mRNA-based anti-rabies vaccines for humans (by Héla Kallel from Univercells). Based on our results and expertise, we consider that we are not able to include this type of statement in our manuscript.

We hope that an article on RNA-based anti-rabies vaccines will be submitted by other researchers and included in this special issue of Viruses, as they represent an interesting alternative.
